# Photoredox cobalt-catalyzed regio-, diastereo- and enantioselective propargylation of aldehydes via propargyl radicals

Lei Wang[1], Chuiyi Lin[1], Qinglei Chong[1] ✉, Zhihan Zhang[2] ✉ & Fanke Meng [1,3,4] ✉

Catalytic enantioselective introduction of a propargyl group constitutes one of the most important carbon–carbon forming reactions, as it is versatile to be transformed into diverse functional groups and frequently used in the synthesis of natural products and biologically active molecules. Stereoconvergent transformations of racemic propargyl precursors to a single enantiomer of products via propargyl radicals represent a powerful strategy and provide new reactivity. However, only few Cu- or Ni-catalyzed protocols have been developed with limited reaction modes. Herein, a photoredox/cobalt-catalyzed regio-, diastereo- and enantioselective propargyl addition to aldehydes via propargyl radicals is presented, enabling construction of a broad scope of homopropargyl alcohols that are otherwise difficult to access in high efficiency and stereoselectivity from racemic propargyl carbonates. Mechanistic studies and DFT calculations provided evidence for the involvement of propargyl radicals, the origin of the stereoconvergent process and the stereochemical models.

Development of efficient and selective catalytic processes to access enantioenriched homopropargyl alcohols are highly desirable due to their frequent utility in the synthesis of natural products and biologically active molecules[1]. Pioneering investigations have led to the development of protocols for addition of stoichiometric amounts of enantiomerically enriched allenylmetal compounds (Sn-, Zn-, B-, Si-, or In-based) to aldehydes that provide access to homopropargyl alcohols with high diastereoselectivity[1,2]. Groundbreaking studies have identified chiral catalysts for additions of Sn-[3–6], Zn-[7], Si-[8–11] or B-based[12–16] allenyl or propargyl reagents to aldehydes (Fig. 1a). However, significant limitations remained unsolved. In the majority of the transformations above, only a simple propargyl group was incorporated; in

few cases, an additional propargylic methyl-substituted stereogenic center was generated[17]. Pre-formation of stoichiometric amounts of organometallic reagents was required. Cr-Catalyzed enantioselective propargylation of aldehydes[18–20] and ketones[21] has been revealed and only a simple propargyl group was able to be introduced. Stoichiometric amounts of reactants and reagents for cleavage of the Cr–O bond to enhance catalyst turnover were required. A more compelling strategy entails in situ generation of allenyl–metal intermediates through enantioselective metal–hydride[22–26] or metal–B(pin)[27] addition to 1,3-enynes followed by diastereoselective addition to aldehydes (Fig. 1b). Although the initial preparation of an organometallic reagent was obviated, the diversity of propargyl groups and the variety of

[1]State Key Laboratory of Organometallic Chemistry, Center for Excellence in Molecular Synthesis, Shanghai Institute of Organic Chemistry, University of Chinese Academy of Sciences, 345 Lingling Road, 200032 Shanghai, China. [2]CCNU-uOttawa Joint Research Center, Key Laboratory of Pesticide & Chemical Biology, Ministry of Education, College of Chemistry, Central China Normal University, 152 Louyu Road, Wuhan 430079 Hubei, China. [3]State Key Laboratory of Elemento-Organic Chemistry, Nankai University, Tianjin, China. [4]School of Chemistry and Materials Science, Hangzhou Institute for Advanced Study, University of Chinese Academy of Sciences, 1 Sub-lane Xiangshan, 310024 Hangzhou, China. ✉e-mail: chongql@sioc.ac.cn; zhihanzhang@ccnu.edu.cn; mengf@sioc.ac.cn

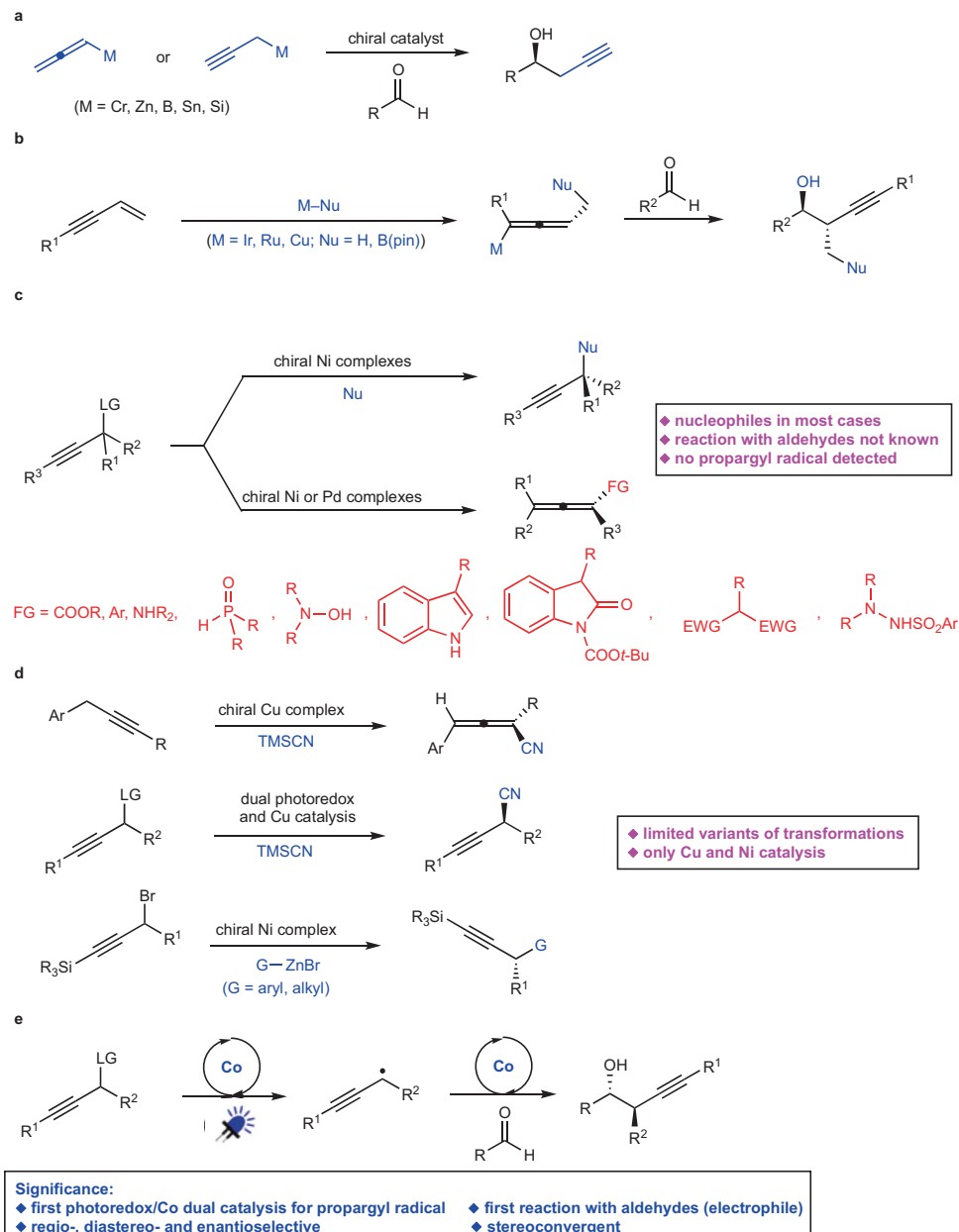

**Fig. 1 | Catalytic enantioselective propargylation and reaction design.**
**a** Catalytic enantioselective propargylation of aldehydes with stoichiometric amounts of organometallic reagents. **b** Catalytic enantioselective propargylation of aldehydes initiated by metal–nucleophile addition to 1,3-enynes. **c** Catalytic enantioselective reactions of racemic propargylic precursors via chiral allenyl–metal intermediates. **d** Catalytic enantioselective transformations via propargyl radicals. **e** Catalytic diastereo- and enantioselective propargyl addition of aldehydes via propargyl radicals (*this work*).

second propargylic stereogenic centers were limited. For instance, approaches for introduction of a propargyl group bearing an aryl group at the propargylic stereogenic center remained unknown. Despite a few more reports on catalytic enantioselective generation of allenyl–Cu species[28–35], subsequent addition to aldehyde remained rare[27].

Another catalytic enantioselective strategy for generation of chiral allenyl–metal intermediates is oxidative insertion to racemic propargyl precursors. Although recent breakthroughs on catalytic transformations of racemic propargyl precursors to enantioenriched propargyl or allenyl products have been disclosed, only Ni-[36–42] and Pd-based[43–47] catalysts have been applied to such reactions (Fig. 1c). In most cases, a nucleophile reacted with the allenyl–metal intermediate. Transformations of the allenyl–metal species with aldehydes remained undeveloped. Moreover, few examples of enantioselective

propargylation through propargyl C–H functionalization have been developed[48,49]. However, the detection of propargyl radicals was reported in none of the protocols. Tuning the reactivity of propargyl radicals by a metal-based catalyst to enable enantioselective construction of a propargylic stereogenic center has been attracting increasing attentions (Fig. 1d)[50]. Liu et al. reported an enantioselective propargyl C–H cyanation via propargyl radicals promoted by a Cu complex[51]. Subsequently, enantioselective cyanation of propargyl esters induced by dual photoredox and Cu catalysis was developed[52]. Fu et al. revealed Ni-catalyzed enantioselective coupling of racemic propargyl bromides with aryl or alkyl-Zn reagents via propargyl radicals[53,54]. During the submission of this manuscript, Cr-catalyzed enantioselective propargylation of ketones involving propargyl radicals have been revealed[55]. The same group also reported Cr-catalyzed enantioselective allenylation of aldehydes with propargyl halides via

**Fig. 2 | Co-catalyzed propargylation of aldehyde in the absence of a ligand.** The reaction was performed by using CoCl₂ (10 mol %), **1a** (1.5 equiv), **2a** (1.0 equiv) and Mn (2.0 equiv) in MeCN at 22 °C for 12 h.

propargyl radicals[56]. Significant limitations remained unaddressed in this area. Only Cu- and Ni-based catalysts have been disclosed; the reaction modes were restricted to cyanation and Negishi coupling; transformations with aldehydes via propargyl radicals remained unknown.

Cobalt is an inexpensive earth-abundant transition metal of low toxicity[57–59]. Our group has been focusing on developing cobalt-catalyzed enantioselective transformations[60–66]. Recently, we have disclosed a cobalt-catalyzed protocol for diastereo- and enantioselective reductive allyl additions to aldehydes via allyl radicals generated from allylic alcohol derivatives[63]. Such stereoconvergent process entailed oxidative addition of Co(I) complex to allylic alcohol derivatives to form allyl–Co(III) intermediates, which underwent homocleavage to afford allyl radicals followed by capture with the Co(II) species. Racemic starting materials could be converted to a single enantiomer of the homoallyl alcohol products. We imagined to expand such concept to stereoconvergent propargyl addition to aldehydes with racemic propargyl carbonates via propargyl radicals. Compared with allyl addition, it is far more challenging for the propargyl addition. In contrast to the allyl addition that proceeded through a well-defined six-membered transition state, propargyl addition to aldehydes via allenyl–metal species through a distorted cyclic transition state rendered more difficulties to control the regio-, diastereo- and enantioselectivity[1]. The catalyst has to effectively tune the reactivity of the allenyl–Co and the isomeric propargyl–Co intermediate, enabling one of them to react selectively. In addition, to furnish the homopropargyl alcohols in high diastereo- and enantioselectivity, the axial stereogenicity of the allenyl–Co or the selectivity for the stereogenic propargyl–Co intermediate has to be accurately controlled. The same catalyst has to induce high selectivity for the aldehyde addition. When we applied the reaction conditions for our previous allyl addition to aldehydes[63] to transformation of propargyl carbonate **1a** with benzaldehyde **2a**, we found that unlike the allyl addition process, no enantioselective induction was observed for all chiral ligands, and even without a ligand, a mixture of diastereomers of homopropargyl alcohols were obtained (Fig. 2). The loss of enantioselectivity might result from the presence of Mn salt generated from reduction of the Co(II) salt. The Mn salt might be able to activate aldehyde, causing it to react without the need of a ligand. The coordination of Mn salt might lead to the distorted cyclic transition state unfavored, and the reaction might proceed through an opened transition state instead. To circumvent this issue, we thought that organophotoredox catalysis might serve as an appropriate alternative for single electron transfer processes to avoid the presence of additional Lewis acidic metal salts. Enantioselective reactions promoted by photoredox and cobalt dual catalysis have emerged as a powerful tool for the synthesis of chiral compounds[67–71]. However, there is a lack of approaches for cobalt-catalyzed enantioselective transformations via propargyl radicals. Herein, we reported a protocol for regio-, diastereo- and enantioselective propargyl addition to aldehydes with racemic propargyl carbonates via propargyl radicals (Fig. 1e). A broad scope of homopropargyl alcohols with single or a second stereogenic center that are otherwise difficult to access were afforded in high efficiency and stereoselectivity. Mechanistic studies and DFT calculations

disclosed the origin of the stereoconvergent process and the stereochemical models.

## Results and discussion

Our studies commenced with reaction of propargyl carbonate **1a** with benzaldehyde **2a** in the presence of Co complexes derived from various chiral phosphine ligands (Table 1). Unlike our previous protocol for allyl addition[63], the propargyl addition to aldehyde could not be promoted by the Co complex derived from phosphinooxazoline **5a** (entry 1). As expected, it is promising that Co complexes formed from chiral phosphines **5b–d** were able to induce low enantioselectivity (entries 2–4). Using a photoredox catalyst for single-electron transfer got rid of the formation of the additional metal salt, shutting down the possible background reactions. Further investigations indicated that transformation in the presence of electron-rich phosphines provided higher enantioselectivity (entries 6–10). Although reaction promoted by Co complex generated from **5f** delivered homopropargyl alcohol **3a** in 11% yield with 52:48 dr and 63:37 er (entry 6), it was found that ligands containing a benzene backbone and chiral phospholane fragments (**5h–i**) produced not only higher efficiency but also diastereo- and enantioselectivity (entries 8–9). Optimization on solvents revealed that reaction could not proceed in CH₂Cl₂ (entry 11). Lower efficiency was obtained in high polar solvents (entries 12–13). Conducting the reaction in MeCN produced >95:5 dr and 94:6 er with a slight diminishment of yield (entry 14). Reaction in alcoholic solvent afforded racemic product in low efficiency (entry 15). Transformation performed in non-polar solvent provided high stereoselectivity with slight erosion of yield (entry 16). Lowering the Co catalyst loading to 5.0 mol % and loading of the photoredox catalyst to 1.0 mol % resulted in no diminishment of efficiency and stereoselectivity (entry 17). It is worth mentioning that diastereoselectivity was dramatically influenced by the chiral catalyst, suggesting although the propargyl addition to aldehyde proceeded through a six-membered cyclic transition state, unlike allyl addition, the diastereoselectivity was controlled by the catalyst rather than the substrate.

With the optimal conditions in hand, we next investigated the substrate scope for aldehydes (Fig. 3). Aldehydes containing electron-deficient (**3b–f**), electron-rich (**3g–h, 3l–m**), halogenated (**3i–k**), and sterically demanding (**3o–q**) aryl groups are suitable substrates. Boryl group (**3n**) is well tolerated in the reaction. A wide range of heteroaryl aldehydes (**3r–x**) could be converted to the desired homopropargyl alcohols in 68–90% yield with 94:6–98:2 er as a single diastereomer. Reactions of aliphatic aldehydes bearing primary (**3y, 3aa**) and secondary (**3z, 3ab**) alkyls afforded the homopropargyl alcohols in 43–82% yield with 92.5:7.5–95.5:4.5 er as a single diastereomer. There was a match/mismatch effect in the transformations of enantiomerically pure aldehydes. Reaction of aldehyde derived from (S)-lactic acid promoted by Co complex derived from (R,R)-**5h** provided **3ac** in 82:18 dr, whereas a single diastereomer **3ad** was obtained with (S,S)-**5h**. Highly functionalized homopropargyl alcohols (**3ae–af**) containing multiple stereogenic centers were accessed as a single diastereomer by proper choice of the enantiomer of the ligand. α,β-Unsaturated aldehydes cannot participate in the reaction. Only reduction of α,β-Unsaturated aldehydes occurred.

**Table 1 | Optimization of reaction conditions**

Me₃Si—≡—CH(OBoc)(Ph)  **1a** (1.5 equiv) + PhCHO **2a** →
10 mol % CoCl₂, 10 mol % ligand, 2.0 mol % 4CzIPN, 20 mol % i-Pr₂NEt, 1.5 equiv Hantzsch's ester, solvent, blue LEDs, 22 °C, 14 h →
Ph—CH(OH)—CH(Ph)—≡—SiMe₃  **3a**

| Entry | Ligand | Solvent | Yield (%)[a] | dr[b] | er[c] |
|---|---|---|---|---|---|
| 1 | 5a | THF | <5 | NA[d] | NA[d] |
| 2 | 5b | THF | 26 | 56:44 | 45:55 |
| 3 | 5c | THF | 14 | 64:36 | 40:60 |
| 4 | 5d | THF | 11 | 58:42 | 47:53 |
| 5 | 5e | THF | <5 | NA[d] | NA[d] |
| 6 | 5f | THF | 11 | 52:48 | 63:37 |
| 7 | 5g | THF | 32 | 83:17 | 33:67 |
| 8 | 5h | THF | 92 | >95:5 | 97:3 |
| 9 | 5i | THF | 44 | >95:5 | 96:4 |
| 10 | 5j | THF | 17 | 87:13 | 13:87 |
| 11 | 5h | CH₂Cl₂ | <5 | NA[d] | NA[d] |
| 12 | 5h | DMSO | 37 | >95:5 | 62:38 |
| 13 | 5h | DMF | 58 | >95:5 | 94:6 |
| 14 | 5h | MeCN | 70 | >95:5 | 94:6 |
| 15 | 5h | MeOH | 21 | >95:5 | 50:50 |
| 16 | 5h | toluene | 75 | >95:5 | 96:4 |
| 17[e] | 5h | THF | 92 | >95:5 | 97:3 |

**Table 1 (continued) | Optimization of reaction conditions**

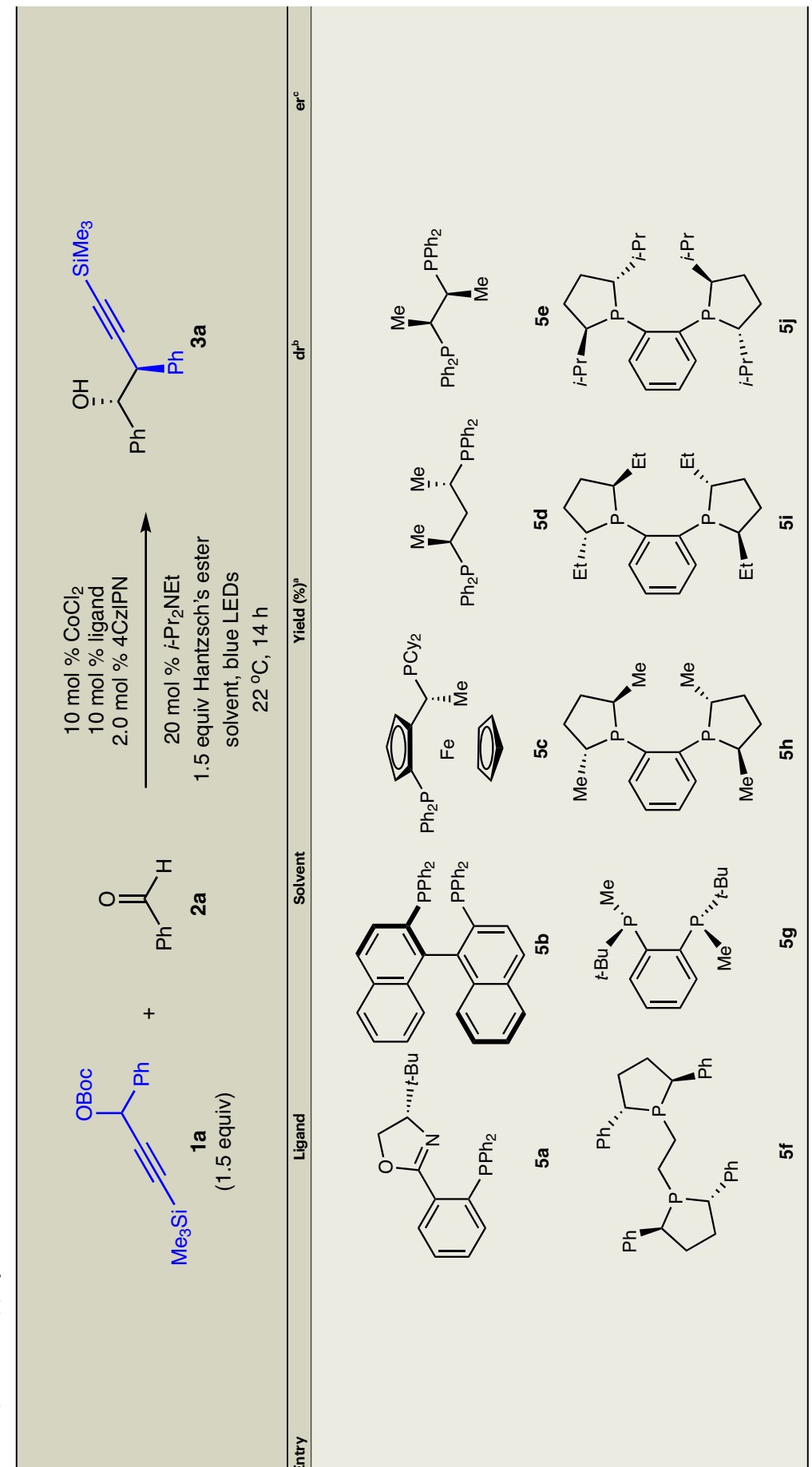

| Entry | Ligand | Solvent | Yield (%)[a] | dr[b] | er[c] |
|-------|--------|---------|--------------|-------|-------|

Reactions were conducted in the presence of CoCl₂ (10 mol %), **5h** (10 mol %), 4CzIPN (2.0 mol %), *i*-Pr₂NEt (20 mol %), **1a** (1.5 equiv), **2a** (1.0 equiv) and 1.5 equiv Hantzsch's ester (1.5 equiv) at 22 °C for 14 h.

[a]Yield of a mixture of diastereomers isolated.

[b]Determined by analysis of 1H NMR spectra of unpurified mixtures.

[c]Determined by analysis of HPLC spectra.

[d]Not available.

[e]The reaction was performed in the presence of 5.0 mol % **5 h**, 5.0 mol % CoCl₂ and 1.0 mol % 4CzIPN. 4CzIPN = 1,2,3,5-tetrakis(carbazole-9-yl)-4,6-dicyanobenzene, Hantzsch's ester = diethyl 1,4-dihydro-2,6-dimethyl-3,5-pyridinedicarboxylate.

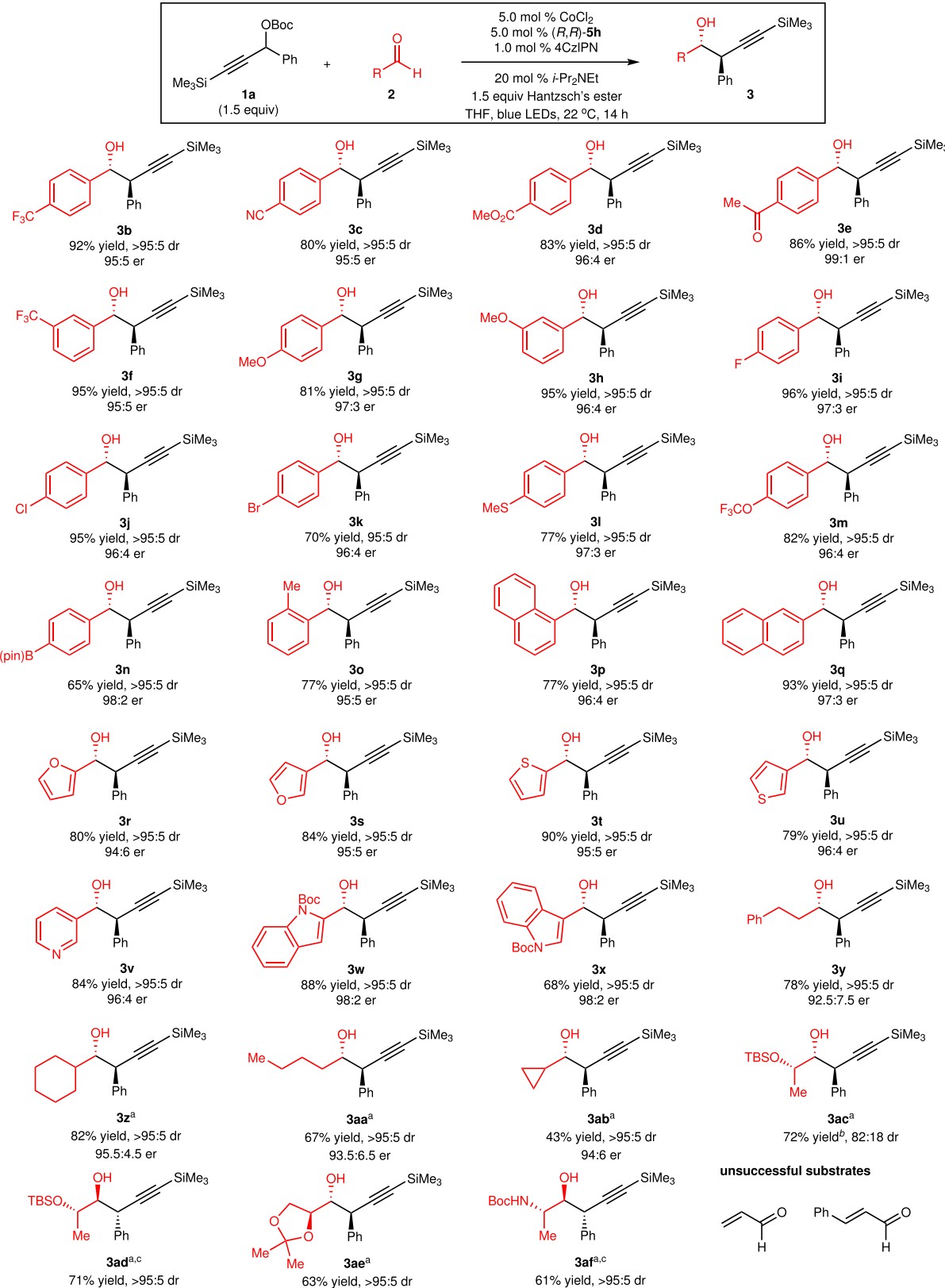

**Fig. 3 | Scope of aldehydes.** All reactions were performed under $N_2$ atmosphere, dr was determined by analysis of [1]H NMR spectra of unpurified mixtures, and er was determined by analysis of HPLC spectra. [a]The reaction was performed in the presence of 2.0 equiv of propargyl carbonate and 2.0 equiv of Hantzsch's ester for 17 h. [b]Yield of pure major diastereomer. [c]The reaction was performed in the presence of 5.0 mol % (S,S)-**5h**. 4CzIPN = 1,2,3,5-tetrakis(carbazole-9-yl)−4,6-dicyanobenzene, Hantzsch's ester = diethyl 1,4-dihydro-2,6-dimethyl-3,5-pyridinedicarboxylate.

Substrate scope extends to propargyl carbonates as well (Fig. 4). A broad scope of racemic propargyl carbonates bearing electron-deficient (6a–c, 6 h), electron-rich (6d, 6i), halogenated (6e–g), and sterically congested (6j–l) aryl groups can participate in the reaction. A variety of heteroaryl-containing homopropargyl alcohols (6m–r) were constructed in 53–80% yield with 95:5–98:2 er as a single diastereomer. Modification of the trimethylsilyl to tert-butyldimethylsilyl group led to an improvement of enantioselectivity (6s). Propargyl carbonates that contain an alkyl-substituted alkyne (6t–w) are well tolerated albeit with lower enantioselectivity. Alkyl-substituted propargyl carbonate was able to be converted to the desired homopropargyl alcohol 6x with 67:33 dr and 95.5:4.5 er. 35% yield of pure major diastereomer was isolated. Homopropargyl alcohols (6y–z, 6aa–ab) bearing a quaternary center can be furnished in 48–91% yield with 88:12–95:5 er. Reaction of primary propargyl carbonate substituted with a silyl group afforded a 69:31 mixture of propargyl and allenyl addition products with 96:4 er for the propargyl addition product (58% yield of pure propargyl addition product 6ac was isolated) and >99.5:0.5 er for the allenyl addition product, whereas exclusive propargyl addition products generated from primary propargyl carbonates containing phenyl groups were obtained in 41–51% yield with 87:13–94:6 er (6ad–aj). Such results suggested that with a larger substituent at the alkyne raised the energy of the allenyl–Co intermediate and lowered the energy of the propargyl–Co intermediate, resulting in a competitive allenyl addition. Propargyl carbonates bearing terminal alkynes and 1,3-enynes were unreactive.

It should be noted that the reaction can be performed on gram scale in a single pot without the need of a continuous flow equipment, affording the enantioenriched homopropargyl alcohol 3a in 92% yield with 97:3 er as a single diastereomer (Fig. 5a). The homopropargyl alcohol bearing two stereogenic centers can be functionalized to a variety of chiral building blocks that are otherwise difficult to access (Fig. 5b). Semi-reduction of the internal alkyne led to exclusive formation of (Z)-homoallyl alcohol 7 in 74% yield without erosion of stereoselectivity[72]. Desilylation of 3a with $K_2CO_3$ in MeOH furnished terminal alkyne 8 in 96% yield. Sonogashira coupling of 8 in the presence of 4.0 mol % $PdCl_2(PPh_3)_2$ and 8.0 mol % CuI afforded 9 in 92% yield with 98:2 er as a single diastereomer, enabling introduction of highly diversified substituents at the alkyne. Click reaction promoted by CuI provided 1,2,3-triazole 10 in 96% yield[73]. Pd-catalyzed conversion of the terminal alkyne in 8 to indole delivered 11 in 92% yield with 97:3 er as a single diastereomer[74]. Transformation of the terminal alkyne moiety in 8 through Cu-catalyzed multicomponent hydrative amidation led to simultaneous cyclization to form γ-lactone 12 in 73% yield[75].

To gain some mechanistic insights, a series of experiments were conducted (Fig. 6). Treatment of propargyl carbonate 1a with a mixture of 2b and 2b-D resulted in a significant inverse secondary isotopic effect (KIE = 0.82), implying that partial rehybridization of the carbonyl carbon from sp² to sp³ and addition of the allenyl–Co species to the aldehyde might be the rate-determining step (Fig. 6a). Monitoring the alteration of enantiomeric ratio of racemic propargyl carbonate 1a recovered from the reaction indicated that both enantiomers of the propargyl carbonate were able to undergo oxidative addition to generate propargyl–Co/allenyl–Co species with a subtle rate difference (Fig. 6b). There is a match/mismatch effect between the chiral catalyst and the stereogenic center of the propargyl carbonate. 80% of the excess amount of the propargyl carbonate 1a was converted to allene 13 with 57:43 er (Fig. 6c), suggesting that both diastereomers of the allenyl–Co species were generated. As only one diastereomer of the product was obtained, dynamic kinetic transformation of the two diastereomers of the allenyl–Co intermediates occurred. Reaction of enantioenriched 1a (80:20 er) in the presence of (R,R)-5h or (S,S)-5h delivered 3a and ent-3a respectively, indicating that the stereochemistry of the product was solely controlled by the chiral catalyst

(Fig. 6d–e). Measuring the enantiomeric ratio of the recovered 1a supported that (R)-enantiomer of 1a reacted faster in the reaction promoted by (S,S)-5h. Transformation of 1a with 2a in the presence of deuterated Hantzsch's ester afforded deuterated allene 13-D, implying that Hantzsch's ester supplied proton for the reaction (Fig. 6f). Control experiments indicated that Hantzsch's ester quenched the excited photocatalyst and served as the reductant (Supplementary Table 3, Supplementary Information). However, in the absence of i-Pr₂NEt, no reaction occurred, suggesting that the oxidized form of Hantzsch's ester cannot proton the O–Co bond directly. i-Pr₂NEt assisted the proton transfer from the reduced form of Hantzsch's ester to protonate the O–Co bond and released the product and stoichiometric amount of i-Pr₂NEt was not required. Trapping the propargyl radical with 5,5-dimethyl-1-pyrroline N-oxide and taking the EPR spectra for the adduct, we detected the presence of a propargyl radical (Fig. 6g, see Supplementary Information for more details). In addition, the adduct was detected in HRMS as well.

Combining all the above mechanistic experiments, we further carried out density functional theory (DFT) calculations to investigate the detailed reaction mechanism and probe the origin of stereoselectivity (Fig. 7). The calculated free energy profile is shown in Fig. 7a. The radical recombination between the Co(II) complex ⁴INT1 and propargyl radical to afford a triplet allenyl-Co(III) complex ³INT2 is slightly exothermic by 0.3 kcal/mol, and the corresponding energy barrier for this radical coupling via ³TS1 was calculated to be only 1.7 kcal/mol. In ³TS1, the computed spin densities of Co and C atoms are 2.597 and −0.395 respectively, indicating a radical coupling process between quartet Co(II) and propargyl radical. The formation of the singlet allenyl-Co(III) complex is highly endothermic by more than 18 kcal/mol, ruling out the possibility of radical coupling between doublet Co(II) complex and propargyl radical. Further single-electron reduction of allenyl-Co(III) to allenyl-Co(II) by 4CzIPN⁻ is exothermic by 2.8 kcal/mol. The eventual asymmetric nucleophilic addition of allenyl-Co(II) ²INT3 on aldehyde 2a requires an activation barrier of 15.0 kcal/mol in which ²TS2 lies higher than the previous allenyl radical capture transition state, establishing itself as the stereo-determining step. This is consistent with the experimental observations that the radical recombination is reversible where both diastereomers of allenyl-Co species were generated. In the meanwhile, the transition state (TS) of nucleophilic addition also witness a partial rehybridization of carbonyl carbon from sp² to sp³, endorsed by the inverse secondary KIE (0.82). To delve into the origin of stereoselectivity, we analyzed the steric environment of the ligand and carefully compared the geometries of nucleophilic addition TSs with different configurations. The topographic steric map[73] shows that the methyl group pointing outward renders quadrant 2 (Q2) more congested, while the less bulky Q3 with the methyl group pointing inside could accommodate more sterically demanding functional group (Fig. 7b). Given that the formation of allenyl-Co(II) ²INT3 is reversible, the stereoselectivity is sorely determined by the interaction between the aldehyde and the chiral allenyl-Co(II) complex. We examined all plausible TSs where four plausible isomers of ²INT3 attack on either Re or Si face of aldehyde (see Supplementary Fig. 8), and Fig. 1c depicts the optimized structures of the most favorable transition states delivering four possible diastereomers respectively. DFT calculations showed that ²TS2top-RR, the TS leading to the experimentally observed product 3a, is the most favored one among the four different configurations. In ²TS2top-RR, the trimethylsilyl (TMS) and the phenyl (Ph) of allenyl group locate on less bulky Q3 and Q1 respectively, matching well with the chiral environment of the ligand. On the other hand, in ²TS2btm-SS, ²TS2btm-RS, and ²TS2top-SR, either the TMS group or the Ph group encounter large steric repulsion with the methyl of ligand in the crowded Q2. These repulsions cause larger dihedral angles Co-C-C-Si in the three disfavored TSs ²TS2btm-SS, ²TS2btm-RS, and ²TS2top-SR and thus weaker coordination of allenyl groups. The additional distortion/

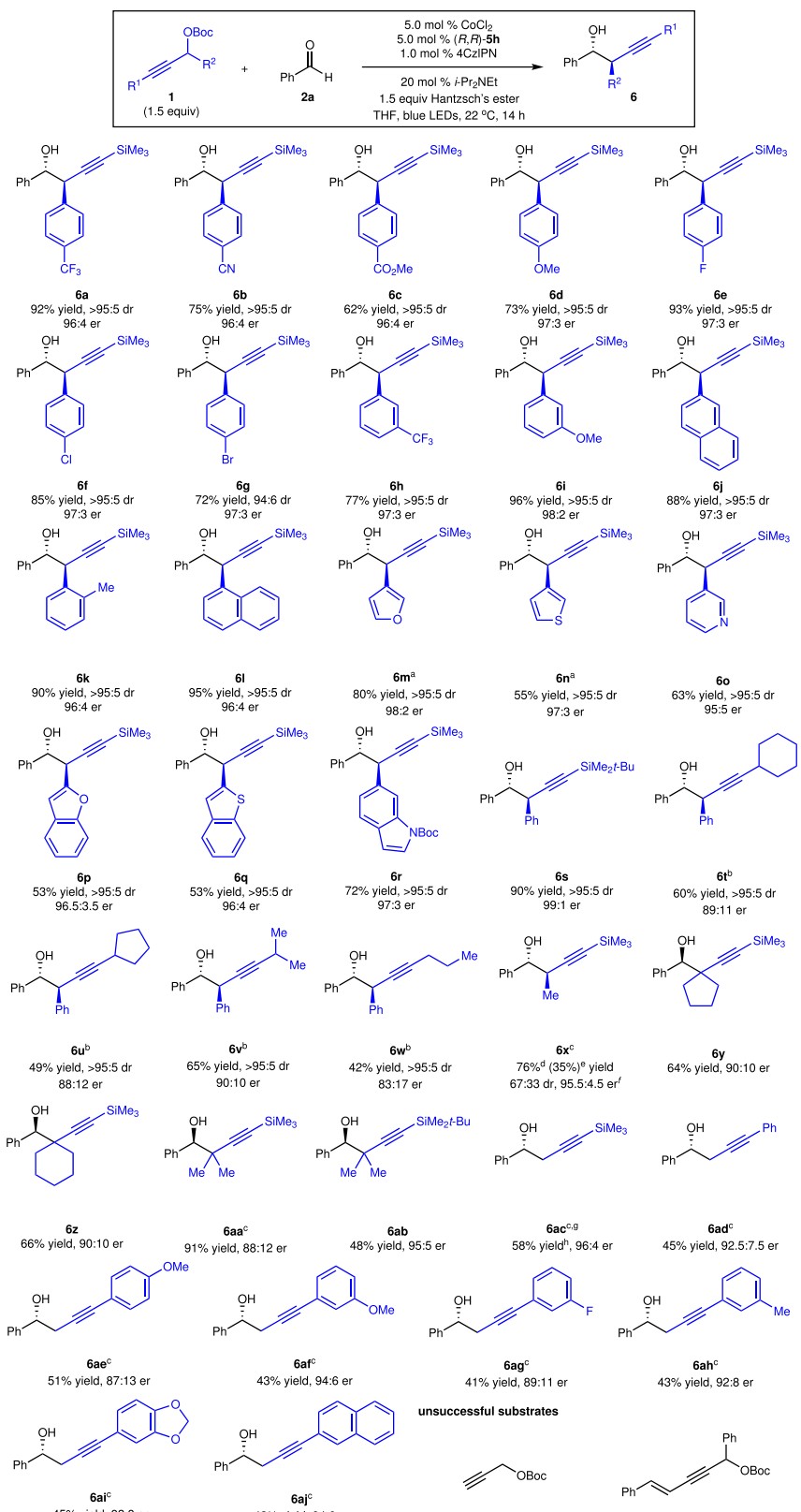

**Fig. 4 | Scope of propargyl carbonates.** All reactions were performed under N₂ atmosphere, dr was determined by analysis of ¹H NMR spectra of unpurified mixtures, and er was determined by analysis of HPLC spectra. [a]The reaction was performed in the presence of 2.0 equiv of propargyl carbonate and 2.0 equiv of Hantzsch's ester for 17 h. [b]The reaction was performed with 10 mol % CoCl₂ and 10 mol % 5h. [c]The reaction was performed in the presence of 10 mol % CoCl₂,

10 mol % 5h, 2.0 mol % of 4CzIPN, 2.0 equiv of i-Pr₂NEt, and 2.0 equiv of Hantzsch's ester. [d]Yield of a mixture of diastereomers. [e]Yield of pure major diastereomer. [f]Enantiomeric ratio (er) of the major diastereomer. [g]A 69:31 mixture of propargyl addition and allenyl addition products were obtained. [h]Yield of pure propargyl addition product. 4CzIPN = 1,2,3,5-tetrakis(carbazole-9-yl)–4,6-dicyanobenzene, Hantzsch's ester = diethyl 1,4-dihydro-2,6-dimethyl-3,5-pyridinedicarboxylate.

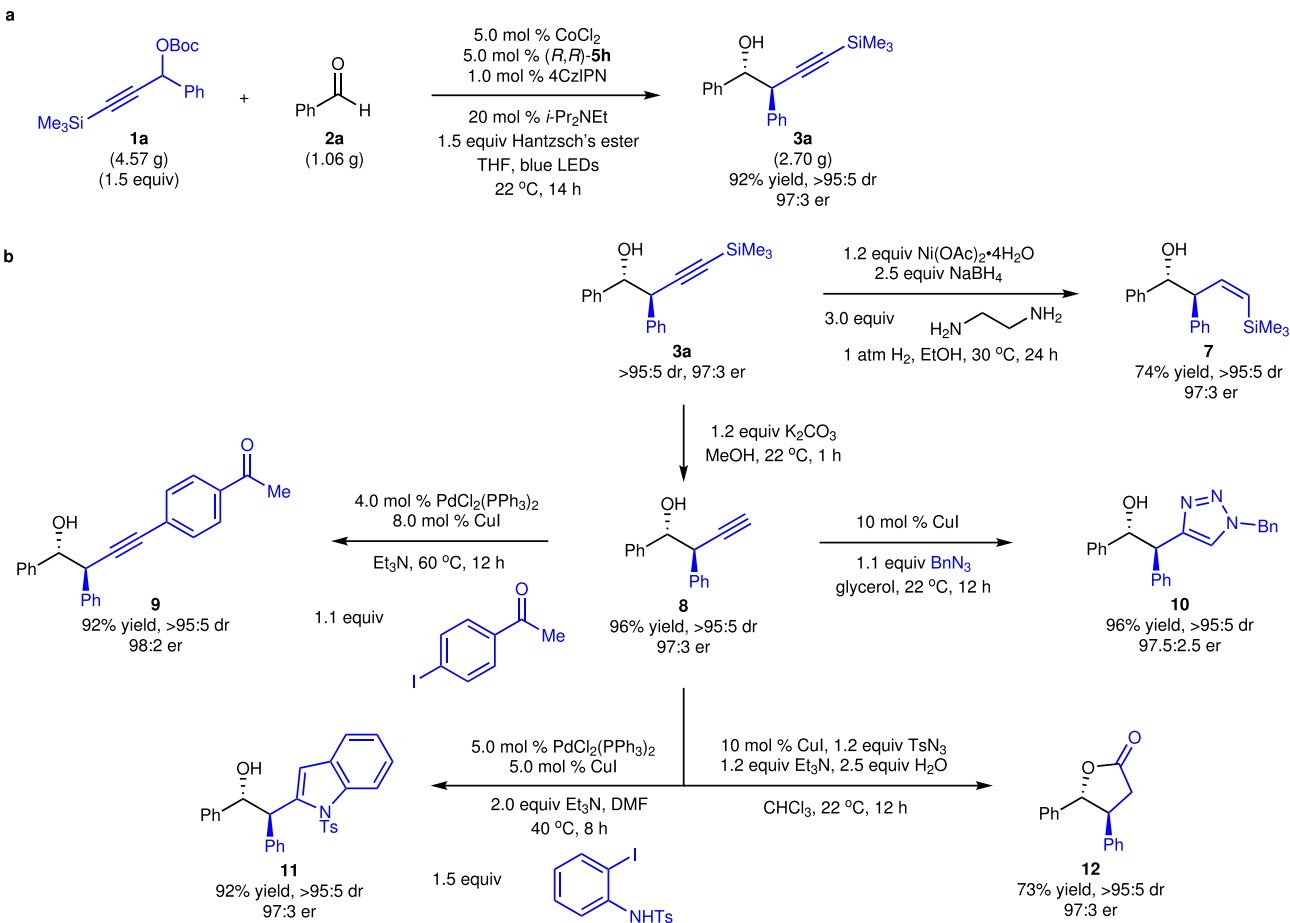

**Fig. 5 | Gram scale reaction and functionalization. a** Gram scale reaction. **b** Functionalization. 4CzIPN = 1,2,3,5-tetrakis(carbazol-9-yl)−4,6-dicyanobenzene, Hantzsch's ester = diethyl 1,4-dihydro-2,6-dimethyl-3,5-pyridinedicarboxylate.

interaction analysis showed that larger distortion energies of allenyl-Co fragments, results of the identified steric repulsion, dominate the destabilization in those disfavored configurations (see Supplementary Table 4). The TSs in quartet state were calculated to be close in energy with doublet TSs, and both spin states are close in energy and share similar stability trend in terms of different configurations (see Supplementary Fig. 8 and Supplementary Table 6). The TSs in doublet were discussed since the origin of stereoselectivity is of primary interests. In summary, the generation of nucleophile is reversible, and the chiral environment of ligand confines the allenyl-Co(II) complex to selectively undergo nucleophilic addition with aldehyde in *RR* configuration, which assures high enantio- and diastereoselectivity.

Based on all the observations above and literature precedence, we proposed a possible catalytic cycle (Fig. 8). Visible light irradiation excited 4CzIPN, which underwent single electron reduction by HE to generate 4CzIPN˙¯. The 4CzIPN˙¯ was able to reduce the Co(II) complex **I** to Co(I) species **II**. Oxidative addition of Co(I) complex **II** to propargyl carbonate **1a** through two-electron process formed propargyl−Co(III) intermediate **III** that could undergo homocleavage to generate propargyl radical **IV** and Co(II) species **I**. Or propargyl radical **IV** associated with Co(II) species **I** was afforded directly through one-electron process. The re-combination of propargyl radical **IV** and Co(II) complex **I** followed by facile single-electron reduction provided two diastereomers of allenyl−Co(II) species (**VII** and **VIII**). There are four possible modes for subsequent aldehyde addition (**IX**−**XII**). The chiral catalyst could accurately control one diastereomer (**IX**) of the allenyl−Co intermediates to react with the aldehyde, delivering high diastereo- and enantioselectivity for the homopropargyl alcohol product. Such dynamic kinetic transformation of the allenyl−Co species resulted in

the stereoconvergent process. Proton transfer released the product and regenerated the catalyst.

In conclusion, a photoassisted cobalt-catalyzed protocol for regio-, diastereo- and enantioselective propargylation of aldehydes to furnish a wide range of homopropargyl alcohols in high efficiency, diastereo-, and enantioselectivity has been developed. To the best of our knowledge, it is the first time that stereoconvergent transformations of racemic propargyl carbonates with aldehydes via propargyl radicals have been achieved. The starting materials are readily accessible or prepared easily. The broad scope of homopropargyl alcohols without the requirement for pre-formation of organometallic reagents enabled the enantioselective incorporation of propargyl groups without substitution at propargyl position and propargyl groups with a tertiary stereogenic center or a quaternary center. The catalyst is derived from an inexpensive sustainable cobalt salt and a commercially available bisphosphine. The synthetic utility was demonstrated by diverse functionalization of the homopropargyl alcohol, affording a series of useful enantioenriched building blocks that are otherwise difficult to access. Mechanistic studies were conducted to elucidate the reaction mechanism, revealing that addition of allenyl−Co species to aldehyde might be the rate-determining step, and dynamic kinetic transformation of the two diastereomers of allenyl−Co intermediate was responsible for the high diastereo- and enantioselectivity. DFT calculations supported the mechanism involving radical recombination, single electron reduction, and rate-determining nucleophilic addition, and the origin of stereoselectivity was uncovered. Such discoveries unveiled a novel reaction pathway for cobalt catalysis and a unique stereoconvergent process for dynamic kinetic transformation of two diastereomeric allenyl−Co species, opening up new

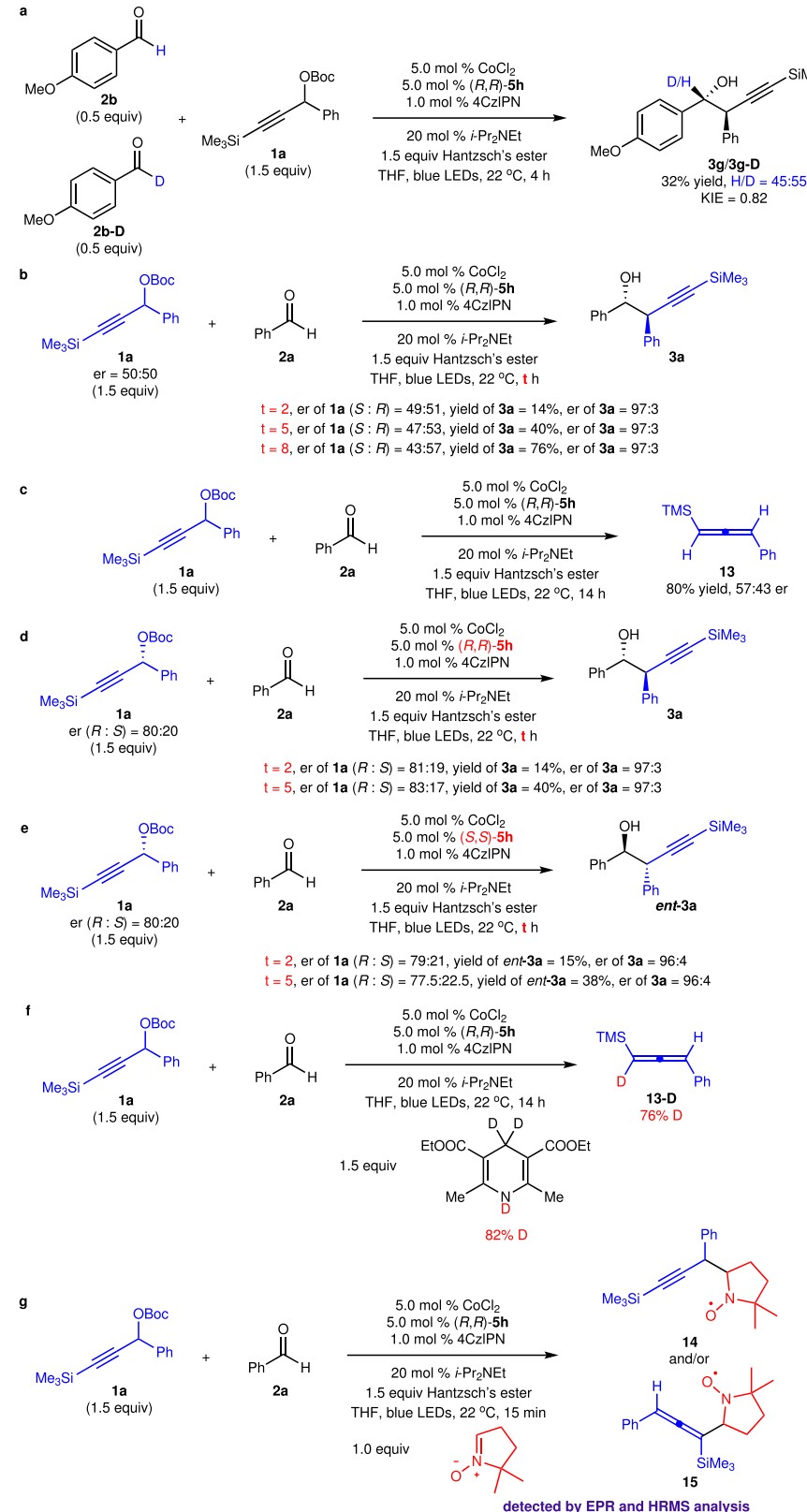

**Fig. 6 | Mechanistic studies. a** Secondary isotopic effect experiments. **b** er of recovered **1a** for reactions of *rac*-**1a** at partial conversion. **c** er of allene **13** generated from protonation of the propargyl–Co intermediate. **d** er of recovered **1a** for reactions of (*R*)−**1a** in the presence of (*R,R*)−**5h** at partial conversion. **e** er of

recovered **1a** for reactions of (*R*)−**1a** in the presence of (*S,S*)−**5h** at partial conversion. **f** Reaction with deuterated Hantzsch's ester. **g** Trapping the radical intermediates. 4CzIPN = 1,2,3,5-tetrakis(carbazole-9-yl)−4,6-dicyanobenzene, Hantzsch's ester = diethyl 1,4-dihydro-2,6-dimethyl-3,5-pyridinedicarboxylate.

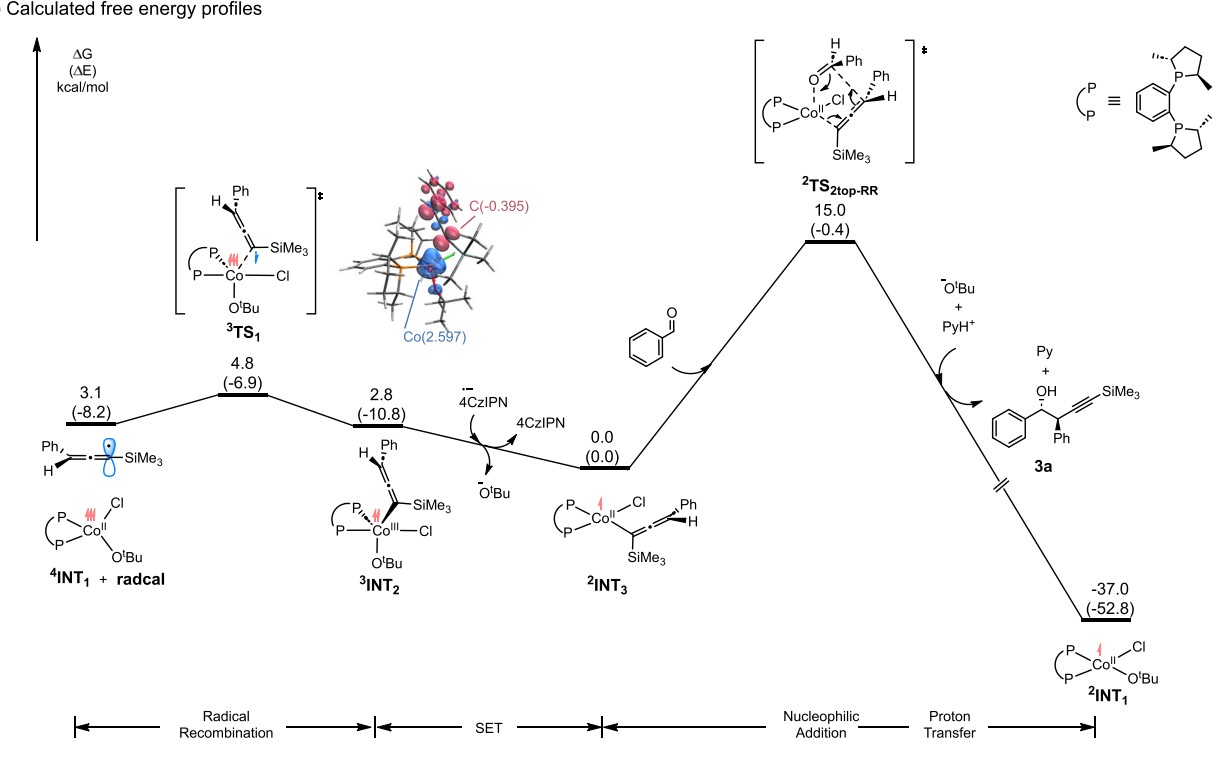

a) Calculated free energy profiles

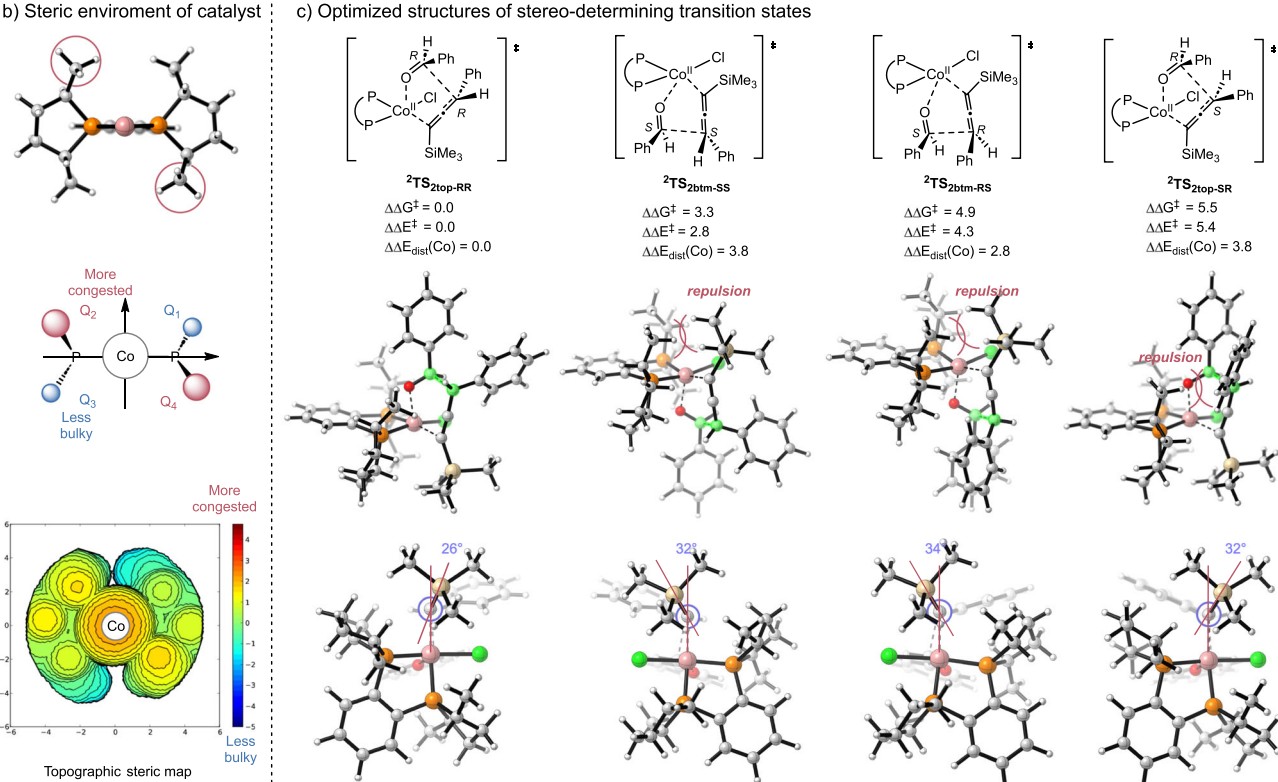

b) Steric enviroment of catalyst

c) Optimized structures of stereo-determining transition states

Topographic steric map

**Fig. 7 | Computational investigations of the reaction mechanism and origin of stereoselectivity. a** Free energy profile calculated at the B3LYP-D3(BJ)/SMD(THF)/def2TZVP//B3LYP-D3(BJ)/def2SVP level of theory (see Supplemental information for more details). **b** Steric environment of catalyst. In the topographic steric map, the area in red means larger steric hindrance while the area in blue means smaller steric hindrance. **c** Optimized structures of stereo-determining transition states. All energies are given in kcal/mol.

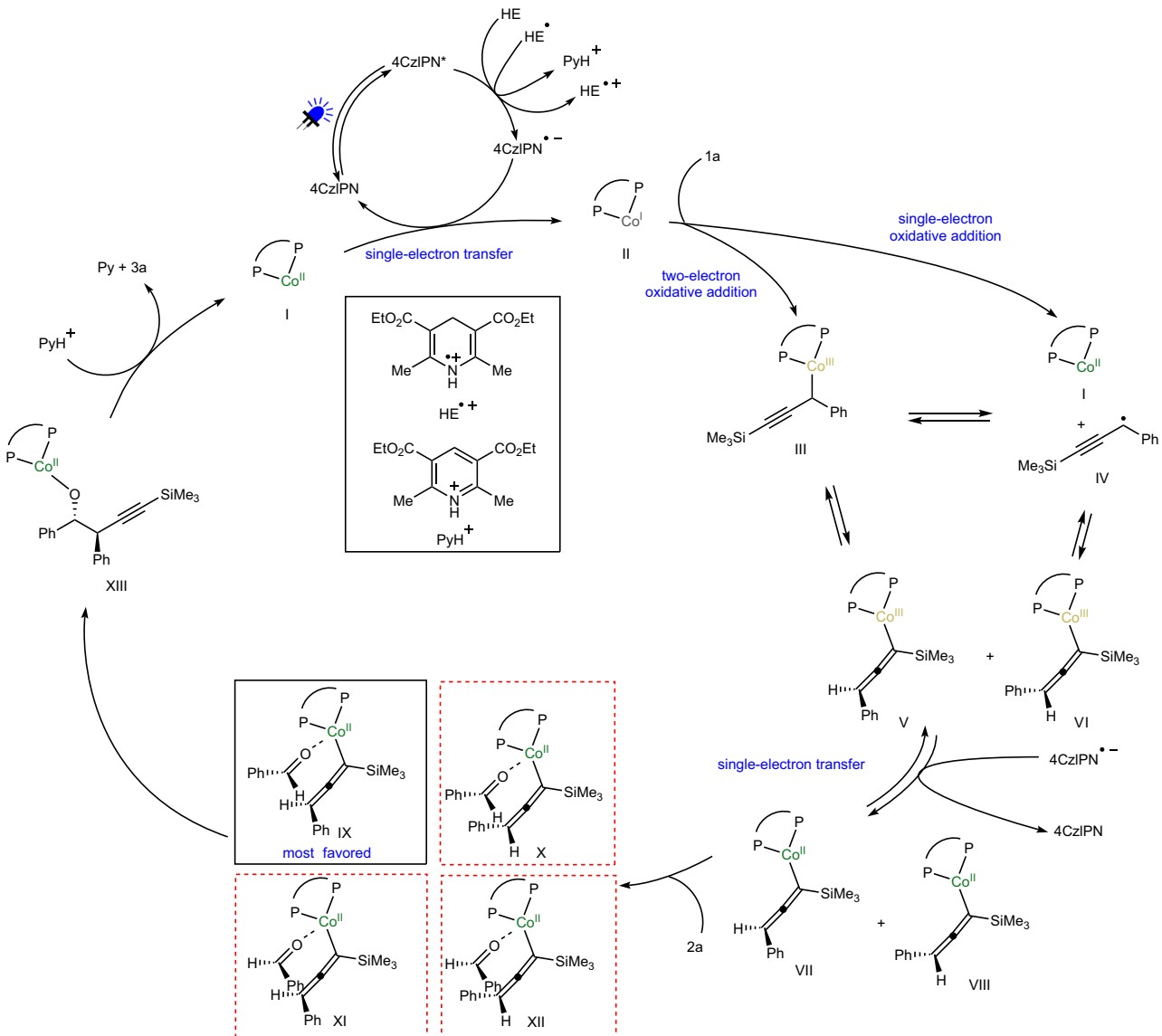

**Fig. 8 | Proposed catalytic cycle.** Pathways are shown for the catalytically reactive species generated from propargyl carbonate and aldehyde. Two diastereomeric allenyl–Co species via radical intermediates were formed reversibly followed by stereoselective addition of one diastereomer to aldehyde effectively controlled by the chiral Co catalyst.

opportunities for designing new reactions promoted by chiral Co-based catalysts and pushing forward the development of organocobalt chemistry. Further investigations on other Co-catalyzed enantiose-lective reactions involving propargyl radicals are underway.

## Methods
### General procedure
In a N$_2$-filled glove-box, an oven-dried vial (8 mL) equipped with a magnetic stir bar was charged with CoCl$_2$ (2.6 mg, 0.02 mmol, 5.0 mol %), (R, R)-**5h** (6.2 mg, 0.02 mmol, 5.0 mol %). THF (3.0 mL) was added, then the mixture was allowed to stir at room temperature for 20 min. **2a** (42.3 mg, 0.4 mmol, 1.0 equiv), **1a** (182.6 mg, 0.6 mmol, 1.5 equiv), i-Pr$_2$NEt (10.3 mg, 0.08 mmol, 0.2 equiv), 4CzIPN (3.1 mg, 0.004 mmol, 1.0 mol %) and Hantzsch's ester (151.9 mg, 0.6 mmol, 1.5 equiv) were added to the solution. The vial was sealed with a cap (phenolic open top cap with red PTFE/white silicone septum) and taken out of the glove box. It was irradiated by 40 W blue LEDs (450-455 nm) and allowed to stir at room temperature (about 22 °C) for 14 h with cooling fans. The mixture is filtered through a short plug of 100–200 mesh silica gel eluting with diethyl ether (3 × 30 mL). The filtrate is concentrated under reduced pressure and the residue was purified by silica-gel column chromatography (eluent: Petroleum ether/ diethyl ether = 12:1) to afford the **3a** as white solid (108.3 mg, 0.37 mmol, 92%).

## Data availability
The authors declare that all other data supporting the findings of this study are available within the paper and Supplementary Information files, and also are available from the corresponding author on request. Crystallographic data generated during this study have been deposited in the Cambridge Crystallographic Data Center (CCDC) under accession number CCDC: 2219864 (**3e**). These data can be obtained free of charge from the CCDC at http://www.ccdc.cam.ac.uk/data_request/cif.

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

## Acknowledgements

We are grateful for financial support from the Nation Key R&D Program of China (2022YFA1506100), National Natural Science Foundation of China (92056103, 21971254, and 21821002), State Key Laboratory of Elemento-Organic Chemistry, Nankai University (202201) and the Shanghai Science and Technology Committee (23ZR1475900).

## Author contributions

Conceptualizaton, F.M.; Methodology, F.M. and L.W.; Investigation, L.W., C.L. and Q.C.; DFT studies, Z.Z.; Writing—original draft, Z.Z and F.M.; Writing—review & editing, F.M., Z.Z, Q.C., C. L., and W.L.; Funding Acquisition, F.M.; Supervision, F.M. and Q.C.

## Competing interests

The authors declare no competing interests.
