## [Peer Review File · Nature Communications]

Reviewers' Comments:

Reviewer #1:

Remarks to the Author:

In this manuscript, Meng et al. report a photo-assisted cobalt-catalyzed regio-, diastereo- and enantioselective propargylation of aldehydes with propargyl carbonates. This reaction provides a photochemical approach to prepare enantioenriched homopropargyl alcohols, which are important building blocks in organic synthesis. Furthermore, a reaction mechanism involving propargyl radicals is proposed and well supported by the experimental results. There has been a plethora of methods to access such a structural motif using different strategies (Ref. 2-34). Particularly, the asymmetric reductive coupling between propargyl electrophiles and aldehydes has been reported (Ref. 8-10). In the current work, propargyl carbonates, cobalt, and Hantzsch's ester replace propargyl halides, chromium, and Mn, respectively. The scope of the propargyl is extended to a variety of secondary and tertiary propargyl groups bearing an internal alkyne moiety while only unsubstituted propargyl is successful in the previous reports. However, the class of reactions is well known, rendering this report an extension of prior work. The authors claim that propargyl radicals are not detected in the previous propargylation. Actually, detailed mechanistic studies in the early contributions (Ref. 8-10) were not conducted. It is probable that these Cr-catalyzed propargylations also involve propargyl radicals, which has been recently confirmed (ACS Catal. 2023, 13, 3170). This discovery deteriorates the mechanistic novelty of the authors' current work. Therefore, this reviewer does not recommend the publication of this manuscript in Nature Communications.

Other comments:

1. Ref. 8-10 do not involve the preformation of stoichiometric organometallics. Instead, readily available propargyl halides are employed directly as the propargylating agent, and propargyl chromium is generated in situ catalytically. Therefore, they should be discussed separately from the Zn-, B-, Sn-, and Si-based propargyl reagents.
2. An important contribution of asymmetric propargylation of aldehydes should be cited: Science, 2011, 333, 1875.
3. Cr-catalyzed enantioconvergent allenylation of aldehydes with racemic propargyl halides (Angew. Chem. Int. Ed. 2022, 61, e202117114) is more related to the authors' work than Ref. 35-53 and thus should be cited.
4. Is this reaction applicable to propargyl carbonates containing a terminal alkyne?
5. Why did the reaction proceed without enantioselectivity in methanol?
6. The authors need to indicate how the dr was determined for compounds 3ad-3af.
7. The authors need to prove the evidence for the Z-geometry of compound 7. The coupling constants of the alkenyl C-H argue for the E-configuration.
8. SI: Full scope of the retention time in HPLC should be shown.
9. SI: For some compounds, the chemical shift -0.0 should be changed to 0.

Reviewer #2:

Remarks to the Author:

Wang and coworkers reported Co/photoredox-catalyzed diastereo and enantioselective propargylation of aldehydes presumably via allenylcobalt intermediate generated from propargyl radical. The authors combined Co diphosphine complex with a reducing photoredox system and realized stereoselective coupling of readily available propargylic alcohol derivatives and mainly aromatic aldehydes. The level of stereoselectivity might not be as excellent as that reported in the propargylation using stoichiometric organometallic reagents, but likely to be good enough for practical synthetic applications. While the catalyst system is in principle similar to that reported in enantioselective allylation (JACS 2021, 143, 12755), in my eyes the optimized conditions reported herein is beyond the logical extension of the previous work and thus deserves positive attention in Nat. Commun. as a methodology study.

However, I think the novelty of the work is not comprehensively supported in the manuscript in the present form. Therefore, to fully satisfy the publication criteria of Nat. Commun. I request the authors to address the following revisions before the manuscript is reconsidered for publication.

- 1) The role of iPr₂NEt should be clarified. This reagent can potentially play several roles in this

reaction: it can serve as an electron donor instead of Hantzsch's ester, or can serve as a ligand on Co instead of the diphosphine. The authors should provide clear explanation why this reagent is needed in 20 mol%, both from the standpoint of reaction design as well as experimental results.

2) Accurate prediction of the spin state of cobalt complexes is still a formidable challenge even using a modern DFT (for example, see *J. Chem. Theory Comput.* 2020, 16, 4416). To validate the theoretical study in the manuscript, the authors should provide a solid experimental benchmark for the current calculation conditions. For example, spin state of INT3 (allenylcobalt intermediate) might be determined to be doublet using EPR.

Reviewer #3:

Remarks to the Author:

Meng, Zhang, and Chong et al developed a photoredox/Cobalt-catalyzed regio-, diastereo- and enantioselective propargyl addition to aldehydes via propargyl radicals. The reaction features with mild reaction conditions, and nice substrate scope, enabling construction of kinds of homopropargyl alcohols in high efficiency and stereoselectivity. Mechanistic studies were well conducted with a combination of experimental and theoretical studies. Thus, the reviewer suggests this paper could be published in *Nature Communications* after solving the following concerns:

1) The manuscript should be carefully checked to remove typos and mistakes. For example, Pls check the caption of Fig. 4 "Scope of aldehyde".

2) About the optimization of reaction conditions, a full optimization table should be included in the supporting information. The referee is curious about the effect of other photocatalysts, 1a/2a ratio, other organic base, etc. Moreover, how can the reaction temperature could be controlled at 22 °C? Has the reaction tube been cooled by water or oil during the reaction? More details should be provided.

3) In the substrate scope, representative unsuccessful cases should be provided. In addition, the referee is curious whether simple aldehyde such as acrolein could be accommodated. Could the Boc group be replaced by other protecting group? For the substrate scope of propargyl carbonates, how about terminal alkynes (replace the TMS group by H) ? Just like 6t and 6aa, more cases with general alkyl or aryl group should be included in Fig. 4, which could be much more useful than the TMS terminal group.

4) For the DFT studies, the referee is curious whether other configurations of Co-complex are considered during the calculations. For example, exchange the position of Cl and allenyl group in TS1 and TS2. The stability of these configurations should be discussed in the SI. Did the author consider the influence of the functional and basis sets used? A comparison with other common functional and basis sets should be included in the SI.

5) More details about the photochemical setup should be provided in the SI to guarantee the peer replication. "LEDs purchased online from Taobao.com" is just like "the LED purchased in the market". This information is useless and makes no sense.

SHANGHAI INSTITUTE OF ORGANIC CHEMISTRY

Fanke Meng
Professor

May 26, 2023

We thank the reviewers very much for a lot of kind and helpful suggestions for improving the paper, inspiring us to think much more for further investigations. The following are point-to-point responses to the requests for revision:

For reviewer 1:

“In this manuscript, Meng et al. report a photo-assisted cobalt-catalyzed regio-, diastereo- and enantioselective propargylation of aldehydes with propargyl carbonates. This reaction provides a photochemical approach to prepare enantioenriched homopropargyl alcohols, which are important building blocks in organic synthesis. Furthermore, a reaction mechanism involving propargyl radicals is proposed and well supported by the experimental results. There has been a plethora of methods to access such a structural motif using different strategies (Ref. 2-34). Particularly, the asymmetric reductive coupling between propargyl electrophiles and aldehydes has been reported (Ref. 8-10).”

Our response: We cannot agree with the reviewer’s comments. Although there are numerous strategies for catalytic enantioselective propargylation of aldehydes, significant limitations still remain unsolved. As we have already explained in the manuscript, only simple unsubstituted propargyl group and propargyl groups bearing a methyl-substituted stereogenic center or gem-dimethyl were able to be introduced (first paragraph in the manuscript). This reviewer also highlighted ref 8-10 regarding Cr-catalyzed enantioselective propargylation of aldehydes. However, in such cases, only a simple propargyl group can be used in the NHK-type reactions. There is no diversity on propargyl bromide.

“In the current work, propargyl carbonates, cobalt, and Hantzsch’s ester replace propargyl halides, chromium, and Mn, respectively. The scope of the propargyl is extended to a variety of secondary and tertiary propargyl groups bearing an internal alkyne moiety while only unsubstituted propargyl is successful in the previous reports.

However, the class of reactions is well known, rendering this report an extension of prior work.”

Our response: We cannot agree with this reviewer’s comments. As we have clearly explained in the first two paragraphs of the manuscript, our new protocol is not a simple extension of previous work. We presented a new strategy by incorporation of photocatalysis and cobalt catalysis to generate umpolung nucleophilic allenyl-cobalt species from propargyl alcohol derivatives that are easily diversified. Such approach for enantioselective propargylation is previously unknown and completely new to cobalt catalysis, solving significant limitations on propargyl scope in the previous work. This is the first example of cobalt-catalyzed enantioselective propargylation of aldehydes.

“The authors claim that propargyl radicals are not detected in the previous propargylation. Actually, detailed mechanistic studies in the early contributions (Ref. 8-10) were not conducted. It is probable that these Cr-catalyzed propargylations also involve propargyl radicals, which has been recently confirmed (ACS Catal. 2023, 13, 3170). This discovery deteriorates the mechanistic novelty of the authors’ current work. Therefore, this reviewer does not recommend the publication of this manuscript in Nature Communications.”

Our response: We cannot agree with the reviewer’s comments. **It is not scientific and precise to say that previous Cr-catalyzed propargylation is mechanistically same as the protocol that the reviewer mentioned, as different chiral ligands and other reaction conditions were used in these reactions.** We can provide an example for cobalt-catalyzed enantioselective allylation of aldehydes, in the two protocols (ACS Catal. 2021, 11, 2992; Angew. Chem. Int. Ed. 2021, 60, 15266), detailed mechanistic experiments indicated no allyl radical was involved, whereas in our paper (J. Am. Chem. Soc. 2021, 143, 12755), the reaction proceeded via allyl radicals. **Moreover, it is more inappropriate to conclude that the Cr-catalyzed enantioselective propargylation of ketones involving propargyl radicals suggested that our Co-catalyzed enantioselective propargylation of aldehydes via propargyl radicals was similar with the Cr-catalyzed protocols. What the reviewer commented is not related to our manuscript, as Cr-catalyzed reaction via propargyl radicals does not mean that catalytic enantioselective propargylation involving propargyl radicals promoted by any other metals is the same. The paper that this reviewer mentioned (ACS Catal. 2023, 13, 3170) is actually not directly related to our manuscript, as the reactions are very different.** First, the ACS Catal. paper is about NHK-type reactions catalyzed by Cr, whereas our method is a dual photo-/Co-catalyzed

approach; Secondly, the ACS Catal. paper is about propargylation of ketones, high diastereoselectivity can be obtained with cyclic ketones and CF₃-containing ketones, whereas ours is about propargylation of aldehydes; Thirdly, the propargyl groups bearing a primary or secondary alkyl-substituted stereogenic center can be introduced in the ACS Catal. paper, there is no aryl- or dialkyl-substituted case; Whereas in our manuscript, high diastereoselectivity can be obtained with both aryl- and aliphatic aldehydes, and propargyl groups without a substitution and containing aryl- and dialkyl-substituted stereogenic centers can be incorporated; Finally, the propargyl nucleophile precursor is propargyl carbonates whereas propargyl chlorides were used in the ACS Catal. paper.

Other comments that the reviewer raised:

1. "Ref. 8-10 do not involve the preformation of stoichiometric organometallics. Instead, readily available propargyl halides are employed directly as the propargylating agent, and propargyl chromium is generated in situ catalytically. Therefore, they should be discussed separately from the Zn-, B-, Sn-, and Si-based propargyl reagents."

Our response: We have revised accordingly (first paragraph in page 2).

2. "An important contribution of asymmetric propargylation of aldehydes should be cited: Science, 2011, 333, 1875."

Our response: We have added the reference (Ref 21).

3. "Cr-catalyzed enantioconvergent allenylation of aldehydes with racemic propargyl halides (Angew. Chem. Int. Ed. 2022, 61, e202117114) is more related to the authors' work than Ref. 35-53 and thus should be cited."

Our response: We have added the reference (Ref 56).

4. "Is this reaction applicable to propargyl carbonates containing a terminal alkyne?"

Our response: No. No reaction occurred with terminal alkynes.

5. "Why did the reaction proceed without enantioselectivity in methanol?"

Our response: We don't know the exact reason. It might be because MeOH can exchange with CoCl₂ to generate Co(OMe)₂, increasing the steric bulk of the metal

center. The reaction might not be able to proceed through a six-membered transition state.

6. "The authors need to indicate how the dr was determined for compounds 3ad-3af."

Our response: dr values were determined by analysis of ¹H NMR spectra of unpurified mixtures. We added the information in Fig 3 and 4.

7. "The authors need to prove the evidence for the Z-geometry of compound 7. The coupling constants of the alkenyl C-H argue for the E-configuration."

Our response: We have performed NOE experiment for compound 7 and prove the Z-configuration of the alkene (Page S214, Supporting Information).

8. "SI: Full scope of the retention time in HPLC should be shown."

Our response: The HPLC traces have been revised accordingly.

9. "SI: For some compounds, the chemical shift -0.0 should be changed to 0."

The H NMR spectra have been corrected.

For reviewer 2

"Wang and coworkers reported Co/photoredox-catalyzed diastereo and enantioselective propargylation of aldehydes presumably via allenylcobalt intermediate generated from propargyl radical. The authors combined Co diphosphine complex with a reducing photoredox system and realized stereoselective coupling of readily available propargylic alcohol derivatives and mainly aromatic aldehydes. The level of stereoselectivity might not be as excellent as that reported in the propargylation using stoichiometric organometallic reagents, but likely to be good enough for practical synthetic applications. While the catalyst system is in principle similar to that reported in enantioselective allylation (JACS 2021, 143, 12755), in my eyes the optimized conditions reported herein is beyond the logical extension of the previous work and thus deserves positive attention in Nat. Commun. as a methodology study.

However, I think the novelty of the work is not comprehensively supported in the manuscript in the present form. Therefore, to fully satisfy the publication criteria of Nat.

Commun. I request the authors to address the following revisions before the manuscript is reconsidered for publication.”

Our response: We thank the reviewer for the insightful comments and constructive comments.

- 1) “The role of *i*-Pr₂NEt should be clarified. This reagent can potentially play several roles in this reaction: it can serve as an electron donor instead of Hantzsch's ester, or can serve as a ligand on Co instead of the diphosphine. The authors should provide clear explanation why this reagent is needed in 20 mol%, both from the standpoint of reaction design as well as experimental results.”

Our response: Control experiments indicated that *i*-Pr₂NEt served as proton shuttle to assist the proton transfer from the reduced form of Hantzsch's ester to the O–Co bond to release the product (Table S3, Supporting Information). So stoichiometric amount of *i*-Pr₂NEt was not required. 20 mol % was the minimum amount that can obtain the high yield. *i*-Pr₂NEt is unlikely to coordinate with the metal center due to the large steric hindrance. The explanation was added to the text on Page 13.

- 2) “Accurate prediction of the spin state of cobalt complexes is still a formidable challenge even using a modern DFT (for example, see J. Chem. Theory Comput. 2020, 16, 4416). To validate the theoretical study in the manuscript, the authors should provide a solid experimental benchmark for the current calculation conditions. For example, spin state of INT3 (allenylcobalt intermediate) might be determined to be doublet using EPR.”

Our response: We express our gratitude to this reviewer for bringing this point to our attentions. We have conducted single point energy calculations with several commonly used functionals to calculate the spin-splitting energies for **IM3** (see **Table S5**, Supporting Information). While there were slight differences in the calculated spin-splitting energies across the different functionals, both doublet and quartet states are close in energy. The results presented in the manuscript were obtained using B3LYP-D3BJ, which closely aligned with the results obtained using MN15L. MN15L is recognized as one of the most suitable functionals for calculations involving multi-spin states. Furthermore, we observed a similar trend in the relative stabilities of different configurations of stereo-determining transition states when using either B3LYP-D3BJ or MN15L. This finding suggests that the choice of functional does not significantly impact the overall stereoselectivity. Since the primary focus of our study

is on elucidating the origin of stereoselectivity, we discussed the doublet transition states calculated using B3LYP-D3BJ in the manuscript. This analysis allowed us to gain insights into how the chiral pocket of the ligand influences the induced stereoselectivities. All these additional calculations suggested by this reviewer have been added to the supporting information to improve the comprehensiveness our theoretical investigations.

Table S5. Benchmark of different density functional for calculating the spin-splitting energy of IM3. Based on the optimized geometry at the level of B3LYP-D3BJ/def2SVP theory, single point energy calculations were performed using def2TZVP basis set together with SMD(THF) accounting for solvation effect.

	B3LYP-D3BJ	MN15L	M06L	TPSSh
$\Delta G(^4\text{IM3}) - \Delta G(^2\text{IM3})$	-0.2	-2.5	3.0	4.5

Table S6. Comparison of calculated free energy difference among different configurations of stereo-determining transition states using B3LYP-D3BJ and MN15L.

	B3LYP-D3BJ	MN15L
$^2\text{TS}_{\text{top-RR}}$	0.0	0.0
$^2\text{TS}_{\text{top-SS}}$	6.2	7.3
$^2\text{TS}_{\text{top-RS}}$	5.9	9.4
$^2\text{TS}_{\text{top-SR}}$	5.5	6.0
$^2\text{TS}_{\text{btm-RR}}$	6.8	8.7
$^2\text{TS}_{\text{btm-SS}}$	3.3	2.0
$^2\text{TS}_{\text{btm-RS}}$	4.9	5.2
$^2\text{TS}_{\text{btm-SR}}$	7.4	9.7
$^4\text{TS}_{\text{top-RR}}$	2.3	5.6
$^4\text{TS}_{\text{top-SS}}$	4.8	11.3
$^4\text{TS}_{\text{top-RS}}$	3.7	9.7
$^4\text{TS}_{\text{top-SR}}$	3.4	5.5
$^4\text{TS}_{\text{btm-RR}}$	5.6	8.4
$^4\text{TS}_{\text{btm-SS}}$	6.0	8.0
$^4\text{TS}_{\text{btm-RS}}$	3.1	1.5
$^4\text{TS}_{\text{btm-SR}}$	6.1	7.3

We have run the EPR experiments, trying to get information for the allenyl-Co

intermediate. However, we were not able to isolate the intermediate due to its instability. In addition, conducting the EPR experiments with the reaction solution cannot obtain any useful information due to the complexity of the reaction mixtures.

For reviewer 3

“Meng, Zhang, and Chong et al developed a photoredox/Cobalt-catalyzed regio-, diastereo- and enantioselective propargyl addition to aldehydes via propargyl radicals. The reaction features with mild reaction conditions, and nice substrate scope, enabling construction of kinds of homopropargyl alcohols in high efficiency and stereoselectivity. Mechanistic studies were well conducted with a combination of experimental and theoretical studies. Thus, the reviewer suggests this paper could be published in Nature Communications after solving the following concerns:”

Our response: We thank the reviewer for insightful comments and bringing a coupling of issues that will significantly improve the manuscript.

- 1) “The manuscript should be carefully checked to remove typos and mistakes. For example, Pls check the caption of Fig. 4 “Scope of aldehyde”.”

Our response: We have checked and revised accordingly.

- 2) “About the optimization of reaction conditions, a full optimization table should be included in the supporting information. The referee is curious about the effect of other photocatalysts, 1a/2a ratio, other organic base, etc. Moreover, how can the reaction temperature could be controlled at 22 °C? Has the reaction tube been cooled by water or oil during the reaction? More details should be provided.”

Our response: We have incorporated all results for the optimization of reaction conditions including other photocatalysts, 1a/2a ratio, other organic bases, the amounts of Hantzsch’s ester and *i*-Pr₂NEt in Supporting Information (**Tables S1-3, Scheme S2**). We ran the reaction in a constant temperature room (20 °C) with a fan to cool the reaction. We did put a thermometer in the flask and indicated the temperature is 22 °C.

- 3) “In the substrate scope, representative unsuccessful cases should be provided. In addition, the referee is curious whether simple aldehyde such as acrolein could be

accommodated. Could the Boc group be replaced by other protecting group? For the substrate scope of propargyl carbonates, how about terminal alkynes (replace the TMS group by H) ? Just like 6t and 6aa, more cases with general alkyl or aryl group should be included in Fig. 4, which could be much more useful than the TMS terminal group.”

Our response: α,β -Unsaturated aldehydes such as acrolein are not tolerated in this reaction. Only reduction of α,β -Unsaturated aldehydes occurred (first paragraph, Page 7). We have put the unsuccessful cases in Fig 3. Propargyl carbonate with Boc group provided the highest yield. Reactions of propargyl alcohol derivatives with other protecting groups proceeded less efficiently (**Scheme S1**, Supporting Information). Terminal alkynes didn't react probably due to the strong affinity of the terminal alkyne to the Co center (Page 9). We did provide more cases for alkyl- (**6u-w**) and aryl-substituted (**6ae-aj**) alkynes (Fig 4).

- 4) “For the DFT studies, the referee is curious whether other configurations of Co-complex are considered during the calculations. For example, exchange the position of Cl and allenyl group in TS1 and TS2. The stability of these configurations should be discussed in the SI. Did the author consider the influence of the functional and basis sets used? A comparison with other common functional and basis sets should be included in the SI.”

Our response: We express our gratitude to this reviewer for bringing this point to our attentions. Exchanging the position of Cl and allenyl group in transition states corresponds to aldehyde approaching from the other side (see **Figure S6**), and the transition states discussed in the manuscript are most favorable one for each configuration. In the meanwhile, several commonly used functional to calculate spin-splitting energy were examined for **IM3** (see **Table S5**). While there were slight differences in the calculated spin-splitting energies across the different functionals, both doublet and quartet states are close in energy. Furthermore, we observed a similar trend in the relative stabilities of different configurations of stereo-determining transition states when using either B3LYP-D3BJ or MN15L. This finding suggests that the choice of functional does not significantly impact the overall stereoselectivity. All these additional calculations suggested by this reviewer have been added to the supporting information to improve the comprehensiveness our theoretical investigations.

Figure S6. Calculated energy difference for all possible nucleophilic addition transition states for both doublet state and quartet state. All energies are given in kcal/mol. The configurations with aldehyde approaching from bottom is the same with exchanging Cl and allenyl group due to C2 symmetry of the ligand.

Table S5. Benchmark of different density functional for calculating the spin-splitting energy of IM3. Based on the optimized geometry at the level of B3LYP-D3BJ/def2SVP theory, single point energy calculations were performed using def2TZVP basis set together with SMD(THF) accounting for solvation effect.

	B3LYP-D3BJ	MN15L	M06L	TPSSh
$\Delta G(^4\text{IM3}) - \Delta G(^2\text{IM3})$	-0.2	-2.5	3.0	4.5

Table S6. Comparison of calculated free energy difference among different configurations of stereo-determining transition states using B3LYP-D3BJ and MN15L.

	B3LYP-D3BJ	MN15L
${}^2\text{TS}_{\text{top-RR}}$	0.0	0.0
${}^2\text{TS}_{\text{top-SS}}$	6.2	7.3
${}^2\text{TS}_{\text{top-RS}}$	5.9	9.4
${}^2\text{TS}_{\text{top-SR}}$	5.5	6.0
${}^2\text{TS}_{\text{btm-RR}}$	6.8	8.7
${}^2\text{TS}_{\text{btm-SS}}$	3.3	2.0
${}^2\text{TS}_{\text{btm-RS}}$	4.9	5.2
${}^2\text{TS}_{\text{btm-SR}}$	7.4	9.7
${}^4\text{TS}_{\text{top-RR}}$	2.3	5.6
${}^4\text{TS}_{\text{top-SS}}$	4.8	11.3
${}^4\text{TS}_{\text{top-RS}}$	3.7	9.7
${}^4\text{TS}_{\text{top-SR}}$	3.4	5.5
${}^4\text{TS}_{\text{btm-RR}}$	5.6	8.4
${}^4\text{TS}_{\text{btm-SS}}$	6.0	8.0
${}^4\text{TS}_{\text{btm-RS}}$	3.1	1.5
${}^4\text{TS}_{\text{btm-SR}}$	6.1	7.3

- 5) “More details about the photochemical setup should be provided in the SI to guarantee the peer replication. “LEDs purchased online from Taobao.com” is just like “the LED purchased in the market”. This information is useless and makes no sense.”

Our response: We have provided the information of the LED in SI (Last line, Page S10).

Please do not hesitate to contact me if you need any additional information regarding this submission.

Sincerely,
Fanke Meng

Reviewers' Comments:

Reviewer #2:

Remarks to the Author:

[Note from the Editor: Reviewer #2 was asked to look also over the concerns of reviewer #1 who was unavailable to assess the revision.]

The authors provided reasonable revisions to my concerns. Publication without change is recommended.

Reviewer #3:

Remarks to the Author:

The mentioned issues have been well solved. The Reviewer supports the publication of this work in Nat. Commun.

SHANGHAI INSTITUTE OF ORGANIC CHEMISTRY

Fanke Meng
Professor

July 22, 2023

For reviewer 1:

“The authors provided reasonable revisions to my concerns. Publication without change is recommended.”

Our response: We thank this reviewer for the comments.

“The mentioned issues have been well solved. The Reviewer supports the publication of this work in Nat. Commun.”

Our response: We thank this reviewer for the comments.

Please do not hesitate to contact me if you need any additional information regarding this submission.

Sincerely,
Fanke Meng